# The Observed Evolution of Arctic Amplification over the Past 45 Years

Mark C. Serreze[1], Elizabeth Cassano[1,2], Alex Crawford[3], John J. Cassano[1,2], and Chen Zhang[1,2]

[1]Cooperative Institute for Research in Environmental Sciences, National Snow and Ice Data Center, University of Colorado Boulder, CO, USA
[2]Department of Atmospheric and Oceanic Sciences, University of Colorado Boulder, Boulder, CO, USA
[3]Centre for Earth Observation Science, Department of Environment and Geography, University of Manitoba, Winnipeg, MB, Canada

*Correspondence to*: Mark Serreze (mark.serreze@colorado.edu)

**Abstract.** To address research gaps in understanding Arctic Amplification, we use data from ERA5, an observational surface temperature dataset, and sea ice concentration to examine the seasonal, spatial and decadal evolution of Arctic 2-meter and lower tropospheric temperatures and lower tropospheric (surface to 850 hPa) static stability over the past 45 years. A Local Amplification Anomaly (LAA) metric is used to examine how spatial patterns of Arctic 2-meter temperature anomalies compare to anomalies for the globe as a whole. Pointing to impacts of seasonally-delayed albedo feedback, growing areas of end-of-summer (September) open water largely co-locate with the strongest positive anomalies of 2-meter temperatures through autumn and winter and their growth through time; small summer trends reflect the effects of a melting sea ice cover. Because of seasonal ice growth, the association between rising 2-meter temperatures and sea ice weakens from autumn into winter, except in the Barents Sea where there have been prominent downward trends in winter ice extent. Imprints of variable atmospheric circulation are prominent in the Arctic temperature evolution. Low-level (surface to 850 hPa) stability over the Arctic increases from autumn through winter, consistent with the greater depth of surface-based atmospheric heating seen in autumn. However, trends towards weaker static stability dominate the Arctic Ocean in autumn and winter, especially over areas of September and wintertime ice loss. Sea ice thinning, leading to increased conductive heat fluxes though the ice, likely also contributes to reduced stability.

**Non-technical Summary**

The outsized warming of the Arctic relative to the globe as a whole (Arctic Amplification) is largest in autumn and winter, consistent with large transfers of energy from growing areas of open water. Impacts of variable atmospheric circulation are also prominent. AA is small in summer due to the melting sea ice cover. Warming penetrates higher into the atmosphere in autumn compared to winter, but trends towards weaker stability could enable deeper heating as AA further evolves.

32

33

## 1 Introduction

Arctic amplification (AA) refers to the observation that, over the last several decades, the rate of increase in surface air temperature over the Arctic region has been larger than for the globe as a whole. As reviewed by Esau et al. (2023), AA is having impacts on Arctic terrestrial and marine ecosystems, permafrost conditions, ice sheets and glaciers as well as human systems. AA was predicted as a consequence of global warming even in the earliest generation of climate models, and was envisioned as far back as the 19th century (Arrhenius, 1896). Various studies have placed the ratio of Arctic to global warming from two to four, with differences relating to the definition of the Arctic region, data used, the time period examined and the season examined (Yu et al., 2021a; Walsh, 2014; Richter Menge and Druckenmiller, 2020; Jansen et al., 2020; AMAP, 2021; Rantanen et al., 2022). Using several observational data sets and defining the Arctic as the region poleward of the Arctic Circle, Rantanen et al. (2022) find a factor of four warming relative to the globe over the period 1979-2021 based on annual mean temperatures. From comparisons with climate models, they conclude that this large ratio is either an extremely unlikely event, or that the models systematically underestimate AA. Zhou et al. (2024) conclude that the externally forced amplification is three-fold, with natural variability explaining the remainder. The Polar Amplification Model Intercomparison Project (PAMIP; Smith et al., 2019) further investigates the causes and consequences of polar amplification using a coordinated set of numerical model experiments, providing valuable insights into the mechanisms driving AA.

Growing spring and summer sea ice loss, leading to more seasonal heat gain in the ocean mixed layer and subsequent upward heat release in autumn and winter - a seasonally-delayed expression of albedo feedback - is widely accepted as a key driver of AA (Perovich et al., 2007; Steele et al., 2008; Serreze et al., 2009; Screen and Simmonds, 2010a,b; Stammerjohn et al., 2012; Stroeve et al., 2014, Dai et al., 2019). However, based on observations and modeling studies, AA is also recognized as involving a suite of connected contributions including changes in atmospheric circulation and poleward energy transport (Graversen and Burtu, 2016: Woods and Caballero, 2016; Henderson et al., 2021; Previdi et al., 2021, Zhang et al., 2025), Planck feedback (Pithan and Mauritsen, 2014), positive lapse rate feedback (Pithan and Mauritsen, 2014; Stuecker et al., 2018; Previdi et al., 2021), changes in ocean heat transport (Beer et al., 2020), changes in autumn cloud cover (Kay and Gettelman, 2009; Wu and Lee, 2012) and even reduced air pollution in Europe (Navarro et al., 2016; Krishnan et al., 2020). Taylor et al. (2022) provide an insightful history of AA science.

However, much remains to be understood about AA, notably the spatial aspects of its observed evolution, seasonal shifts in its expression and evolution, and the vertical structure of AA in the context of changing static stability. Here, using data from the ERA5 reanalysis, surface temperature observations, and satellite-derived sea ice concentration, we focus on understanding the decadal evolution and seasonal/spatial expressions of Arctic temperature anomalies. The local characteristics of AA are

important, as regional variations can produce different remote influences, including midlatitude climate extremes (Zhou et al., 2023). We show how: 1) the pronounced autumn contribution to AA, through which internal energy gained by the upper ocean in spring and summer in growing open water areas is subsequently released back to the atmosphere, decays into winter as sea ice forms (the exception being in the Barents Sea sector, which has seen pronounced winter ice losses; 2) The decadal evolution of AA is modulated by variable spatial expressions of atmospheric circulation;  3) the deeper vertical extent of pronounced temperature anomalies in autumn than winter is consistent with the seasonal increase in static stability from autumn to winter; and 4) reductions in static stability in autumn point toward increasingly deep penetration of surface warming into the troposphere with continued sea ice loss, and potentially greater impacts of AA on altering weather patterns in lower latitudes (Ding et al., 2024).

## 2 Data Sources

Data from the European Centre for Medium-range Weather Forecasts (ECMWF) reanalysis (ERA5; Hersbach et al., 2020) are used for analysis. Monthly temperature (2 m and the significant levels from 1000 to 500 hPa) and surface and latent heat fluxes were used on the 0.25° x 0.25° horizontal grid from 1979-2024. While ERA5 data are available since 1950, fields since 1979, the advent of the modern satellite database for assimilation, are more reliable. ERA5 is chosen because, in various comparisons of (near-) surface parameters throughout the Arctic, ERA5 performs similarly to or better than other global and regional reanalysis products (Graham et al., 2019; Barrett et al., 2020; Renfrew et al., 2021; Crawford et al., 2022). Reliance is placed on trends and anomalies. Anomalies are referenced to the 30-year period 1981-2010, but comparisons are made with different averaging periods. To assess relationships with sea ice conditions, we use the satellite passive microwave records from the National Snow and Ice Data Center. The satellite passive microwave record provides estimates of concentration and extent from October 1978 through the present at 25-km resolution on a polar stereographic grid by combining data from the Nimbus-7 Scanning Multichannel Microwave Radiometer (SMMR, 1979–1987), the Defense Meteorological Satellite Program (DMSP) Special Sensor Microwave/Imager (SSM/I, 1987–2007) and the Special Sensor Microwave Imager/Sounder (SSMIS, 2007-onwards) (Fetterer et al., 2002).

Our results must be viewed within the context of known problems in ERA5, one being a warm bias in 2-meter air temperature over the Arctic (Yu et al., 2021b; Tian et al., 2024). Compared to an extensive set of matching drifting observations, Yu et al. (2021b) found ERA5 to have a mean bias of 2.34 ± 3.22 °C in 2-meter air temperature, largest in April and smallest in September. Interestingly, surface (skin) temperature biases were found to be negative (−4.11 ± 3.92 °C overall, largest in December and smaller in the warmer months), although the magnitudes might be overestimated by the location of the surface temperature sensors on the buoys, which may have been affected by snow cover. While we are largely dealing in this paper with anomalies, rather than absolute values, our comparisons between Arctic and global anomalies may be influenced by the fact that biases at the global scale are different. Wang et al. (2019) found that compared to the earlier ERA-I effort, ERA5 has

95 a larger warm bias at very low temperatures (< -25°C) but a smaller bias at higher temperatures. ERA5 has higher total
96 precipitation and snowfall over Arctic sea ice. The snowpack in ERA5 results in less heat loss to the atmosphere and hence
thinner ice at the end of the growth season, despite the warm bias.

To further address biases in ERA5, analysis was also performed using the Berkeley Earth Surface Temperatures (BEST)
gridded surface temperature data (Rohde and Hausfather, 2020; Available for download from: https://berkeleyearth.org/data/).
This dataset extends back to 1850, combining both 2m temperatures over land as well as sea surface temperatures to create a
global, gridded observational dataset to which reanalysis data can be compared.

**3 Results**

**3.1 Seasonality of 2-Meter Temperature Trends**

A key, but in our view, under-appreciated aspect of AA is its strong seasonality - under-appreciated not that it exists but in the
sense that processes at work during summer over the Arctic Ocean, when AA is small, set the stage for understanding the
strong imprints of AA during autumn and winter. Rantanen et al. (2022) found that the AA factor as assessed for the region
poleward of the Arctic circle ranges from less than 2 in July to over 5 in November. Climate models examined in that study
largely capture this seasonality but with smaller amplification factors. Figure 1 shows spatial patterns of surface air temperature
trends by season based on ERA5. In this study, the Arctic is defined as areas poleward of 60°N, but maps extend down to $50^{\circ}$N
to enable comparisons between changes in the Arctic and the higher middle latitudes. The same analysis but performed with
the BEST data are shown in Supplemental Figure 1. The description of the results from these figures apply to both datasets
except where explicitly stated.

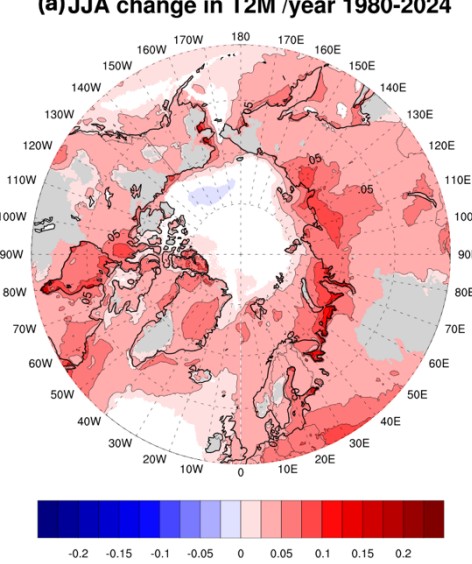

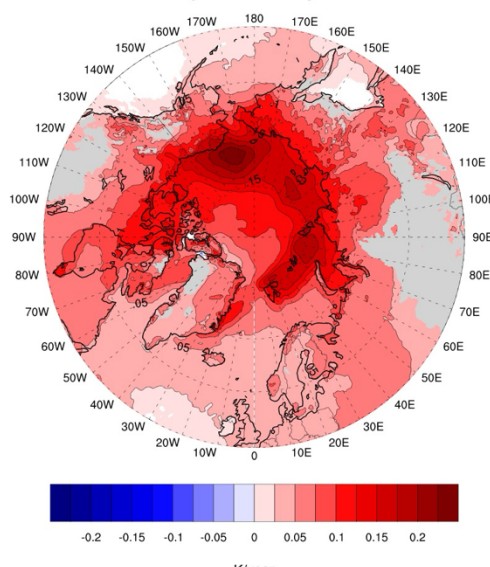

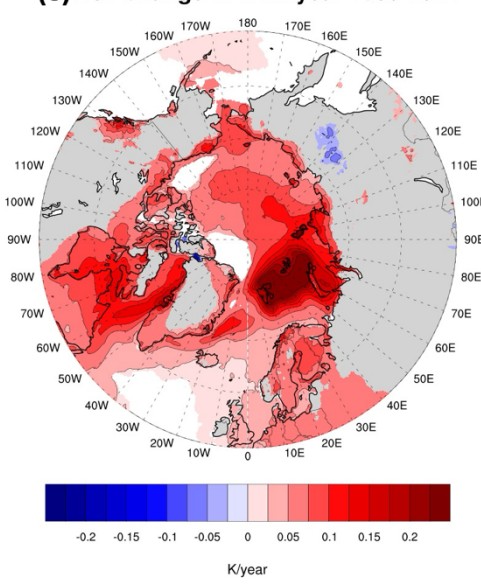

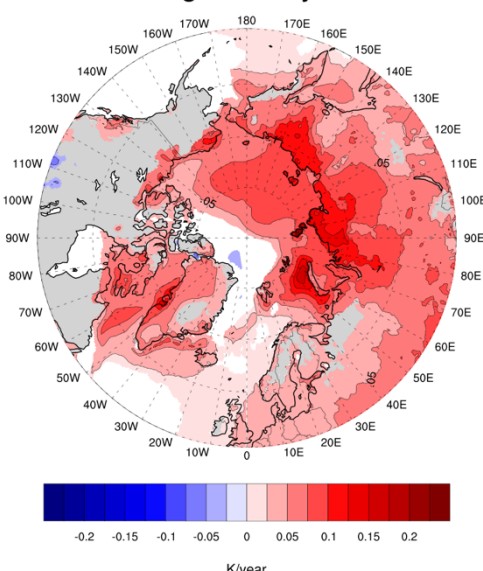

**Figure 1: Linear trends in ERA5 2-meter temperatures (T2M) by season from 1980 to 2024, in degrees per year for (a) June, July,**
**August (JJA), (b) September, October, November (SON), (c) December, January, February (DJF) and (d) March, April, May**
**(MAM). Only trends significant at p<0.05 are shaded based on an ordinary least squares regression test.**

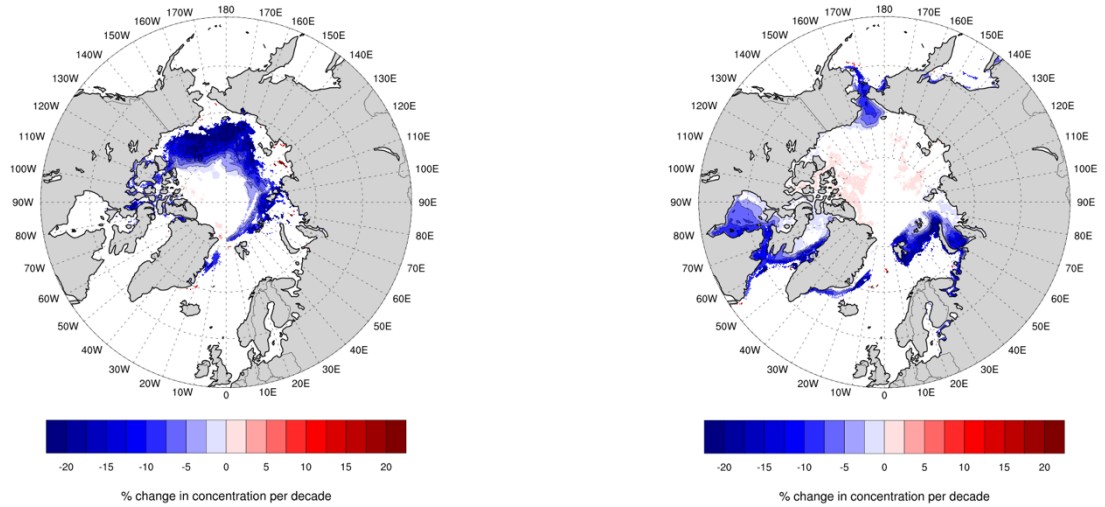

**(a) Sep percent change in seaice concentration/decade 1980-2024**   **(b) Dec percent change in seaice concentration/decade 1980-2024**

**Figure 2: Linear trends in sea ice concentration %/per decade 1980 through 2024 for September (a) and December (b). Only trends significant at p<0.05 are shaded based on an ordinary least squares regression test.**

The sharply smaller trends in summer compared to autumn and winter across Arctic latitudes clearly stands out. In interpreting these patterns, we focus on broad, contiguous regions rather than isolated grid points that may be affected by spatial autocorrelation. Summer trends are nevertheless largely positive and statistically significant across most of the Arctic and subarctic lands. Trends in ERA5 are very small and not statistically significant across the central Arctic Ocean, while in the BEST data, the trends over the Arctic Ocean are significant, albeit still small (Figure S1a). Since the skin temperature of a melting sea ice cover is pegged to the melting point, it follows that surface air temperature trends must be small in this area. Over land, earlier loss of the snow cover (Mudryk et al., 2023) likely contributes to the rise in surface air temperatures seen there. Trends along the Russian and Alaska coastline are also positive. Melt onset typically starts in June in the southern margins of the ice cover and progresses poleward (Markus et al., 2009). Positive trends along the coastal seas are consistent with satellite observations of a progressively earlier onset of melt (Stroeve et al., 2014; Stroeve and Notz, 2018). They are also consistent with progressively earlier exposure of dark open water areas, their expanding coverage through time, and associated increased internal energy in the ocean mixed layer (Perovich et al., 2007; Serreze et al., 2009; Perovich and Polashenski, 2012; Stammerjohn et al., 2012; Dai et al., 2019; Li et al., 2021; Bianco et al., 2024). However, the large specific heat of water and the depth of heating (10-30 m) will limit the rise in surface air temperature. Note also the positive trends over the northern North Atlantic, which is ice-free over the entire year. Somewhat larger trends are found over part of the Kara and Barents Seas.

The largest temperature trends for autumn, locally exceeding 0.2°C per year, lie primarily on the Eurasian side of the Arctic Ocean and north of Alaska. A comparison to the spatial pattern of September (end of summer) sea ice concentration (Figure 2), provides an understanding: the trends are largest in those areas with the sharpest downward trends in ice concentration, most notably in the Chukchi and East Siberian Seas and hence where there will be strong upward surface heat fluxes as the ocean loses the internal energy it gained in summer. Our interpretation, building from the above discussion and from earlier studies (e.g., Stammerjohn et al., 2012; Stroeve et al., 2016; Lebrun et al., 2019), is that through the years, ice begins to retreat earlier and earlier in spring and summer, largely from the shores of Alaska and the Russian coast, exposing areas of dark open water, which absorbs solar energy. This means more energy gain in the ocean mixed layer, and over an increasingly large area, with time. As solar radiation declines in autumn, this energy is released upwards to the atmosphere, seen as positive temperature anomalies that grow in magnitude and spatial coverage with time. Before sea ice forms, all of the internal energy gained in summer must be depleted.

The pattern of winter temperature trends is quite different. The positive trends along the Eurasian coastline and in the Chukchi and Barents Seas are greatly reduced, and the largest trends, exceeding 0.2°C per year, are now located in the Barents Sea. The reason for this is clear: by December, the areas of open water along the coast have re-frozen, reducing energy transfer between the ocean and atmosphere. The Barents Sea is, in turn, one of the few areas with a substantial downward trend in winter sea ice extent (Figure 2b). Still, positive 2-meter temperature trends in both autumn and winter encompass much of the Arctic Ocean away from areas of ice loss. One likely driver of this is progressive thinning of the ice cover (Landy et al., 2022; Sumata et al., 2023), allowing for an increase in conductive fluxes through the ice (Liu and Zhang, 2025). Autumn and winter trends in sensible and latent heat fluxes from ERA5 show an increase over the time period of study of these fluxes from the surface to the atmosphere (Supplemental Figure S2). Another driver is likely polar temperature advection from the areas of sea ice loss (Timmermans et al., 2018), as evidenced by the tongue of fairly large positive trends extending from the Barents Sea into the Arctic Ocean. Also of interest is that trends over much of the land area are very small, even negative, especially over Eurasia.

By spring, the magnitude of temperature trends in both the ERA5 and BEST data over the Barents Sea has dropped relative to winter, but is still prominent. Through spring, downward trends in sea ice concentration (not shown) persist, but, compared to winter, air-sea temperature differences are smaller, hence ocean to atmosphere surface heat fluxes are smaller. Substantial positive trends are found along the Eurasian coast, again suggestive of the role of atmospheric heat advection. Trends over much of high-latitude North America are small.

To summarize, it is apparent that an assessment of Arctic Amplification based on comparing the Arctic trend with the trend for the globe as a whole must recognize the highly pronounced seasonal and spatial heterogeneity of Arctic trends. Summer 2-m temperature trends are mostly small, but the smallness over the Arctic Ocean is due to the melting of ice. The much larger

autumn trends reflect energy transfer from the ocean to the atmosphere via upward surface heat fluxes from increasing extensive areas of open water. By winter, open water areas along the Eurasian coast and the Chukchi Sea have re-frozen and the locus of maximum temperature trends is shifted to the Barents Seas, consistent with the downward trends in sea ice concentration there. Spring trends are weaker than winter trends, but are still large in the Barents Sea sector. However, for autumn, winter and spring, there are also features in the spatial patterns of trends that point to advection and other processes, and winter trends in particular are small over much of the land area.

**3.2 Local Amplification Anomaly Approach**

To gain further insight into trends, we now look at the evolution of AA by decade, 1980-1989, 1990-1999, 2000-2009, and 2010-2019, as well as the last five years of the record, 2020-2024, making use of what we term a Local Amplification Anomaly (LAA) approach.

For each of these periods, we calculated the average 2-meter temperature at each ERA5 and BEST grid point across the globe, then calculated the anomalies at each grid point relative to the 1981-2010 climatology. Taking the (spatially weighted) average of all grid point anomalies yields the global temperature anomaly for each period. Then, at each grid point we subtracted this global temperature anomaly from the anomaly at that point. We then compiled maps of the anomalies for the region poleward of 50°N (including the Arctic (north of 60°N) and the sub-Arctic (50-60°N)). Examining these LAAs gives us a sense of the spatial structure of Arctic temperature anomalies in terms of how they contribute to the overall AA evolution. In Table 1 we also provide, for each decade and season, the average of the anomalies relative to the global average poleward of 60°N and the average global anomaly. Results that follow will of course reflect the chosen 1981-2010 referencing period.

| | Global Anomaly (K) | | Arctic Anomaly (K) | | Difference (Arctic – Global; K) | |
|---|---|---|---|---|---|---|
| Autumn | BEST | ERA5 | BEST | ERA5 | BEST | ERA5 |
| 1980-1989 | -0.22 | -0.22 | -0.76 | -0.74 | -0.54 | -0.52 |
| 1990-1999 | -0.05 | -0.06 | -0.35 | -0.45 | -0.30 | -0.39 |
| 2000-2019 | 0.22 | 0.22 | 0.83 | 0.91 | 0.61 | 0.69 |
| 2010-2019 | 0.42 | 0.45 | 1.51 | 1.68 | 1.09 | 1.23 |
| 2020-2024 | 0.69 | 0.78 | 2.08 | 2.42 | 1.39 | 1.64 |
| Winter | | | | | | |
| 1980-1989 | -0.10 | -0.16 | -0.47 | -0.24 | -0.37 | -0.08 |
| 1990-1999 | -0.02 | -0.03 | -0.56 | -0.53 | -0.54 | -0.50 |
| 2000-2009 | 0.15 | 0.16 | 0.73 | 0.71 | 0.58 | 0.55 |
| 2010-2019 | 0.35 | 0.38 | 1.66 | 1.66 | 1.31 | 1.28 |
| 2020-2024 | 0.54 | 0.62 | 1.35 | 1.38 | 0.81 | 0.76 |
| Spring | | | | | | |
| 1980-1989 | -0.20 | -0.14 | -0.83 | -0.68 | -0.63 | -0.54 |

| | | | | | | |
|---|---|---|---|---|---|---|
| 1990-1999 | -0.01 | -0.04 | 0.23 | 0.13 | 0.24 | 0.17 |
| 2000-2009 | 0.16 | 0.14 | 0.36 | 0.36 | 0.20 | 0.22 |
| 2010-2019 | 0.40 | 0.40 | 1.40 | 1.37 | 1.00 | 0.97 |
| 2020-2024 | 0.58 | 0.60 | 1.37 | 1.16 | 0.79 | 0.56 |
| Summer | | | | | | |
| 1980-1989 | -0.18 | -0.15 | -0.34 | -0.29 | 0.16 | -0.14 |
| 1990-1999 | -0.001 | -0.01 | -0.09 | -0.09 | 0.091 | -0.08 |
| 2000-2009 | 0.14 | 0.13 | 0.33 | 0.28 | 0.19 | 0.15 |
| 2010-2019 | 0.34 | 0.35 | 0.64 | 0.70 | 0.30 | 0.35 |
| 2020-2024 | 0.61 | 0.63 | 0.86 | 1.04 | 0.25 | 0.41 |

**Table 1: Average temperature anomalies (K; with respect to 1981-2010) for the Arctic (north of 60°N), the globe, and their difference for the BEST and ERA5 data.**

Results for autumn are examined first (Figure 3 (ERA5) and Supplemental Figure 3 (BEST data)). The description of the results apply to both datasets unless indicated otherwise. For the first two decades, 1980-1989 and 1990-1999, both the average global anomaly and the average Arctic anomaly are small and negative, with the Arctic anomalies actually more negative than the global value. Since 1980-1989 is (primarily) the first decade of the 1981-2010 baseline period, greater negative anomalies for the Arctic than the globe still indicate amplified warming in the Arctic. Likewise, as the middle of the baseline period, 1990-1999 experiences the smallest anomalies. This pattern reverses starting in the 2000-2009 decade. What this is capturing is that early in the record, the poleward gradient in 2-meter temperatures was stronger than it is today; as AA evolves, the gradient obviously weakens.

For the first decade, 1980-1989, LAAs are generally small across the Arctic, with a mix of positive and negative values, but with the negative anomalies obviously dominating (not shown). The exception is in the Chukchi Sea, where strong negative LAA values of up to 3°C are found. Based on data from 1979-1996, Parkinson et al. (1999) showed downward trends in ice concentration in the Chukchi Sea of around 4% per decade. However, as the area had more sea ice in the 1980-1989 decade relative to the 1981-2010 climatology, it shows up as negative LAA values in Figure 3.As noted, in the 1990-1999 decade, both the Arctic average and the global average anomaly are at their minimum, since this decade is in the middle of the 1981-2010 baseline (Table 1). However, the difference between the 1990-1999 and the subsequent 2000-2009 decade is striking. Both the Arctic and global average anomalies are positive (Table 1, Figures 3 and S3). Positive LAA values encompass most of the Arctic. The largest positive LAA values lie in the Chukchi and East Siberian Seas, reflecting the continuing development through this decade of extensive open waters in September (Figures 3 and S3). Note that the first clear indication of the emergence of AA related to sea ice loss was based on data extending through the end of the 2000-2009 decade (Serreze at al., 2009; Screen et al., 2010a, b). Wang et al. (2017) similarly found the emergence of amplified temperature anomalies over the Arctic (60-90°N) compared to the northern mid-latitudes (30-60°N) in this decade. By the 2010-2019 decade, autumn LAA values of 3-5°C in the ERA5 data (2-4°C in the BEST data) are now prominent along the entire Eurasian coast and in the

Chukchi Sea; consistent with the continued increase in open water areas in September. Much smaller AA values encompass
most of the rest of the Arctic.

The most recent period, 2020-2024, sees a shift. While strongly positive anomalies relative to global average anomalies - that
is, positive LAA values - remain over much of the Eurasian coastal sea, LAA anomalies over the Chukchi Sea are now smaller,
and larger values have appeared in the Beaufort Sea and the Canadian Arctic Archipelago. In explanation, when Arctic sea ice
extent began to decline, it was initially most prominent in the Chukchi Sea region, so LAA values there are especially large,
as seen in the 2009-2009 and 2010-2019 plots. With the rise in the global temperature anomalies, these LAA values become
more subdued.

The winter evolution is quite different. The Arctic-averaged anomaly and the global anomaly for the 1980-1989 are small and
quite alike – AA had not yet emerged (Table 1). In terms of the LAA structure (not shown), positive values of typically 1-2°C
over much of Eurasia, Alaska and Canada contrast with negative values of similar size elsewhere, the exception being negative
values of 2-3°C in the Barents Sea sector. The story is similar for the 1990-1999 decade - AA had yet to clearly emerge (Table
1), and, indeed, the Arctic average anomaly was about half a degree colder than the global average anomaly. The LAA structure
leading to this interesting finding is characterized by partly offsetting positive and negative values (Figure 4 (ERA5) and
Supplemental Figure 4 (BEST data)). As was the case for the discussion of the autumn AA, the description of the results
applies to both datasets unless indicated otherwise. Of interest in this regard is that North Atlantic Oscillation (or Arctic
Oscillation) shifted from a negative to a strongly positive index phase between the 1970s and late 1990s. Numerous studies
examined the strong temperature trends associated with this shift, notably warming over northern Eurasia, with cooling over
northeastern Canada and Greenland (e.g., Hurrell, 1995; 1996; Thompson and Wallace, 1998). There was vibrant debate over
whether the shift might be in part a result of greenhouse gas forcing and an emerging signal of expected Arctic Amplification
(see the review in Serreze et al., 2000).

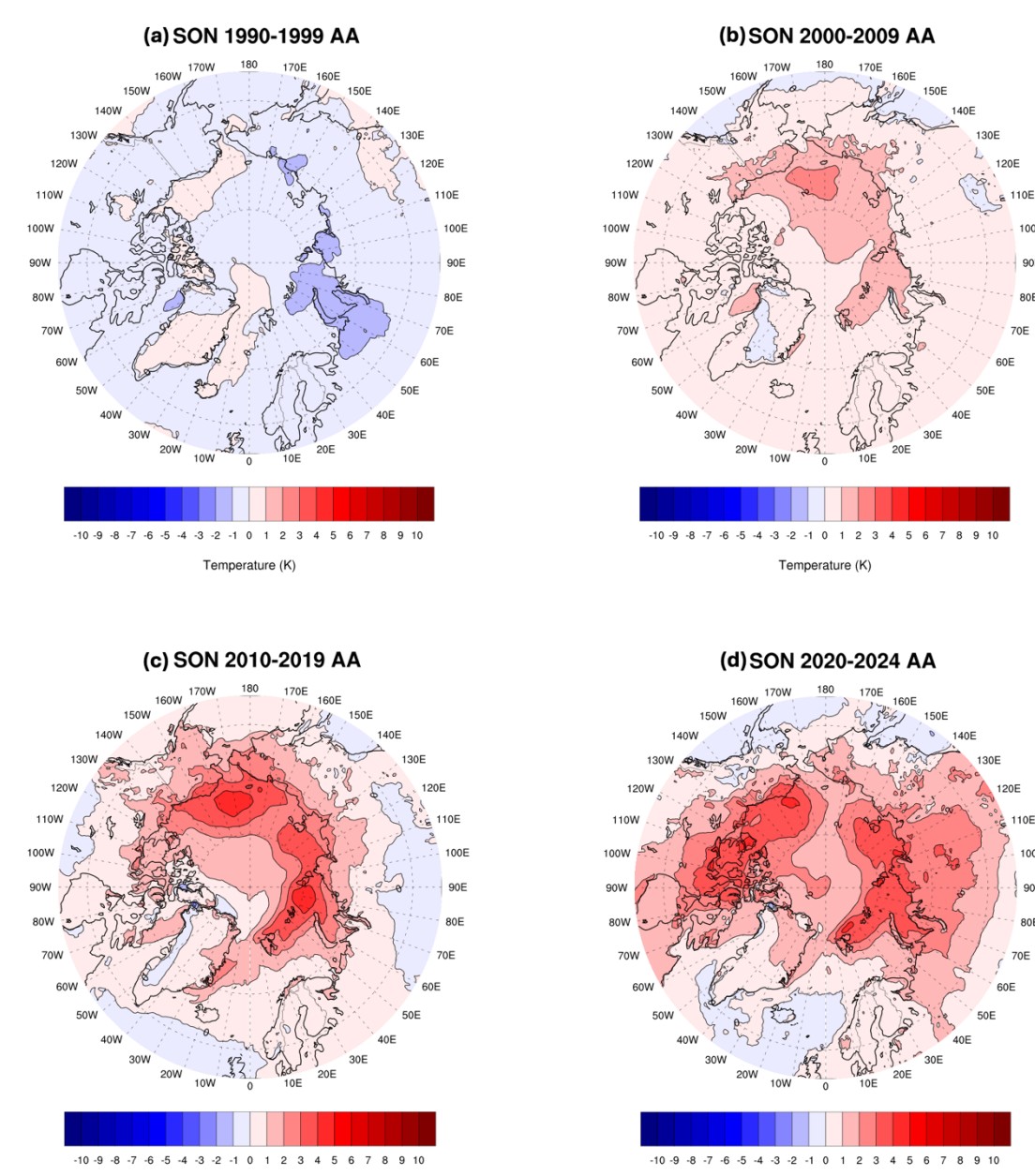

**Figure 3. Autumn (September, October, November (SON)) ERA5 2-m temperature anomalies in ºC relative to 1981-2010 for (a)**
**1990-1999, (b) 2000-2009, (c) 2010-2019 and (d) 2020-2024 minus the global average temperature anomaly for each period.**

While there is some indication of a structure in LAA values for the 1990-1999 decade reminiscent of the rising phase of the NAO over this time (note that the index value subsequently decreased), looking back to Table 1, the behavior of the NAO clearly did not "boost" any emerging AA signal.

Turning to the decade 2000-2009, positive LAA values have become more dominant, and fairly large positive values have appeared over the Barents Sea sector, replacing the negative values of the previous decade. While by this decade, AA had clearly emerged (Table 1), note that the positive LAA values over northern Eurasia in 1990-1999 are replaced by negative values, indicative of a circulation shift, notably, regression of the NAO from its previous high index values.

The 2010-2019 period is characterized by the emergence of large positive LAA values over the Barents Sea sector which have intensified since the 2000-2009 decade, pointing to the effects of growing open water areas in this sector. Positive LAA values also cover almost all Arctic latitudes. The Barents Sea feature remains prominent in the past five years of the record (2020-2024). Note, however, the negative anomalies over Alaska and eastern Eurasia. As a result, the difference between the Arctic average temperature anomaly and the global average anomaly is actually smaller than in the 2010-2019 period, that is, pan-Arctic AA is somewhat smaller. Note also by comparison with the decade 2010-2019, LAA values along most of the Eurasia coast are less pronounced. This is understood in that, by December, all areas along the Eurasian coast and north of the Chukchi and East Siberian seas have refrozen.

The observation that the last three time periods have negative LAA values over Eurasia is of interest through its apparent link with Warm Arctic-Cold Eurasia (WACE) phenomenon -while AA has become increasingly prominent, this has been attended by recent surface cooling over Eurasia, most evident in winter with considerable decadal variability. (e.g., Gong et al., 2017; Li et al. 2021). The WACE phenomenon has garnered considerable attention over the past decades and a suite of driving factors have been offered. An Urals blocking pattern has been identified as playing a strong role, and recent work has shown that decadal variability in the WACE phenomenon is mediated by phases of the Pacific Oscillation and the Atlantic Multidecadal Oscillation (e.g., Luo et al., 2022).

Turning back to the Barents Sea sector, it is notable that this is one of the few areas of the Arctic (along with eastern Hudson Bay/Hudson Strait and Bering Strait, see Figure 2) with substantial downward trends in winter sea ice concentration. Various studies have attributed the loss of winter ice in the Barents Sea and associated temperature anomalies and trends to processes involving atmospheric circulation, facilitating intrusions of warm moist air into the region with wind patterns promoting stronger transport of warm Atlantic waters into the region (Woods and Caballero, 2016; Lien et al., 2017; Siew et al., 2024). Warm and moist air advection raises temperatures, inhibits autumn and winter sea ice growth (Woods and Caballero, 2016; Crawford et al., 2025; Lee et al., 2017), and enhances spring and summer ice melt (Kapsch et al., 2013; Park et al., 2015). Intrusions of Atlantic-derived waters, which appear to be in part wind driven, also discourage winter ice growth. Beer et al.

(2020) identified an oceanic mechanism that increases the vertical heat flux in the upper Arctic Ocean under global warming

that causes increased ocean heat transport into the Arctic, which appears as a substantial contributor to Arctic Amplification.

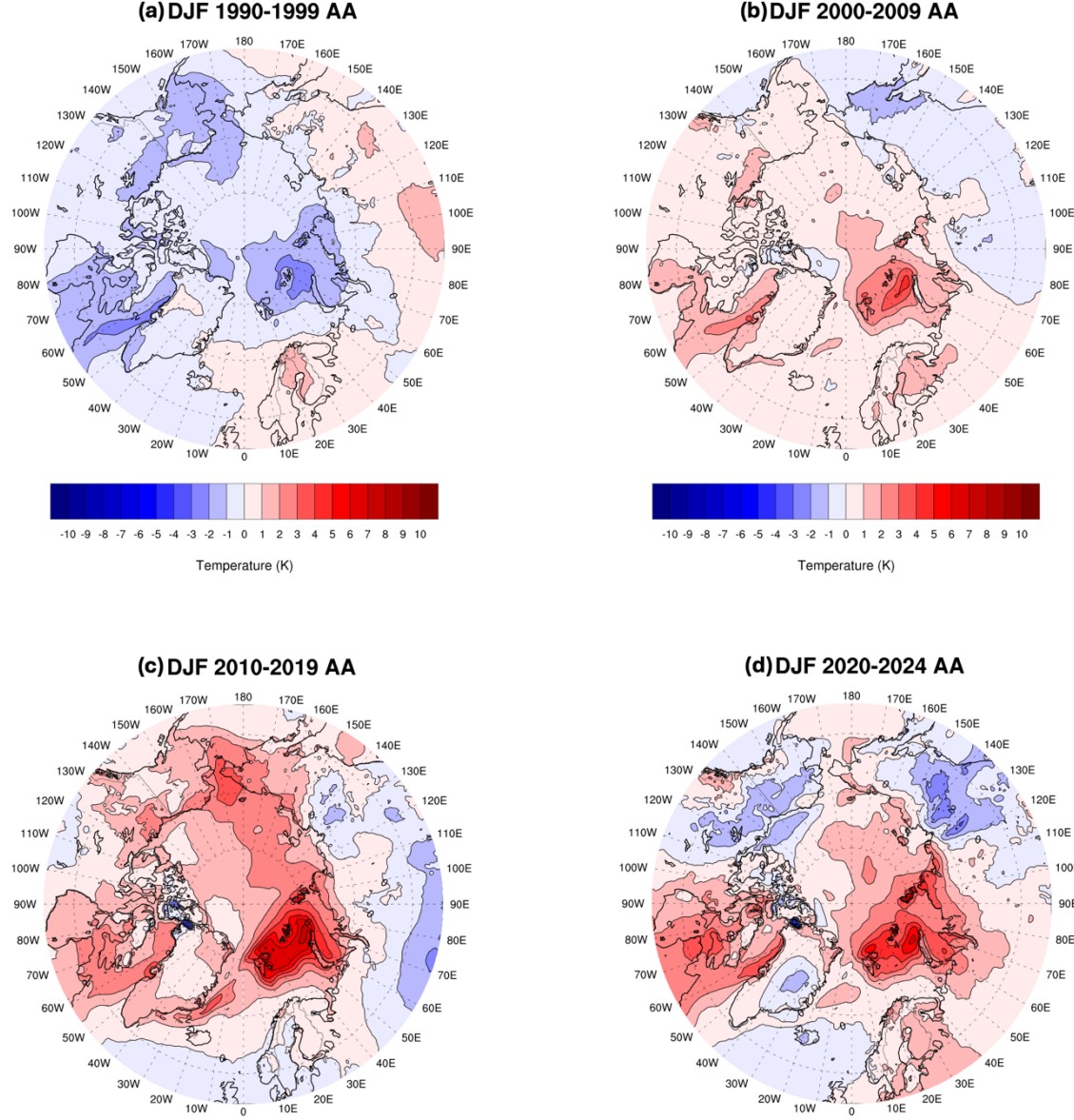

**Figure 4: Winter (December, January, February, DJF) surface temperature anomalies in ºC relative to 1981-2010 for (a) 1990-1999, (b) 2000-2009, (c) 2010-2019 and (d) 2020-2024 minus the global average temperature anomaly for each period.**

While our primary focus is on the evolution of AA and LAAs in autumn and winter, it is warranted to briefly discuss spring and summer (not shown). The spring pattern of LAAs for the 1980-1989 decade is characterized by small and mostly negative values across the Arctic, transitioning to a mix between small positive and negative values for the 1990-1999 decade, as well as for the 2000-2010 decade. The largest difference between the Arctic average and global average anomaly was for the 2010-2019 decade. This is consistent with the much smaller AA in this season compared to autumn and winter. Only for the last five years of the record, 2020-2024 do prominent positive LAA values of over 3°C appear over Eurasia, but these are partly balanced by negative LAAs elsewhere and may represent short-term internal variability. The key feature of summer is that while as the decades pass, modest positive values of LAA appear over land, values remain close to zero over the Arctic Ocean, reflecting the effects of the melting sea ice surface. The last five years also show positive LAA values of up to 3°C along the shores of Eurasia, likely due to the open coastal waters in these areas.

The results just discussed are with reference to 1981-2010 averages. Use of an earlier climatology (e.g., 1951-1980) naturally yields stronger positive anomalies and weaker negative LAA values in the later part of the temperature records, while a more recent climatology (e.g., 1991-2020, the current NOAA standard) has the opposite effect. The 1981-2010 reference applied in this paper is an appropriate middle ground, and is the reference period used for sea ice analyses by the National Snow and Ice Data Center (Scott, 2022).

### 3.3 Vertical Structure

An assessment of the vertical structure of warming helps to both highlight the effects of sea ice and shed light on other processes known to be involved in Arctic Amplification, notably, static stability. To this end, we look at longitudinal cross sections of temperature anomalies for the most recent 10 years of the record, averaged between the latitudes 75-80°N, which corresponds to the latitude band with pronounced anomalies in surface air temperature across both SON and DJF. We look first at October, then turn attention to December (Figure 5). October is when there will be particularly large heat fluxes from the ocean to atmosphere, while in December, most of these areas (apart from the Barents Sea) have re-frozen. This choice of months is intended to capture that contrast.

The strongly positive anomalies located from 60-120°E and between 180°E to 120°W (these being stronger) are clearly surface-based, which makes sense as they are due to strong upward surface heat fluxes. The more prominent feature between 180°E and 120°W (centered along the East Siberian and Chukchi Seas) is notable in that anomalies of 3°C extend up to 700hPa. The December cross section shows maximum surface-based temperature anomalies focused between about 20-70°E (centered near the Barents Sea), but positive anomalies do not extend as far in the vertical compared to October. Although these

anomalies are less vertically extensive, the stronger near-surface temperature difference between the surface and the air above in December could potentially enhance surface fluxes.

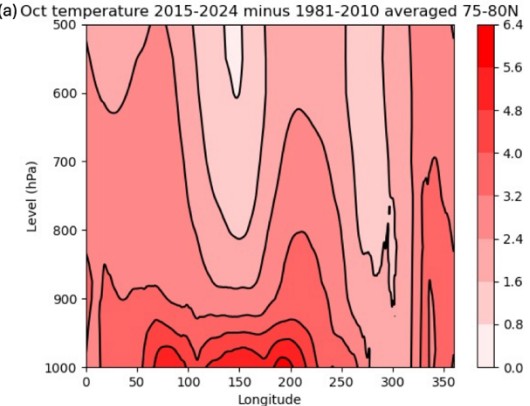 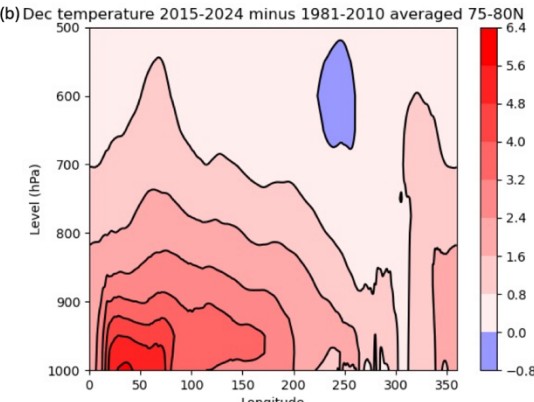

**Figure 5: Vertical cross sections by longitude across latitudes 75°N to 80°N for October (a) and December (b) of temperature anomalies for 2015-2024 minus 1981-2010.**

**3.4 Static Stability**

While the magnitude of the surface temperature anomaly will bear on how high in the vertical positive anomalies will persist, the vertical stability will play a role. The strong stability of the lower Arctic troposphere has long been recognized (Wexler, 1936; Bradley et al., 1992; Kahl et al., 1992; Serreze et al., 1992) and is central to arguments that lapse rate feedback is a contributor to AA. Based on radiosonde observations, Serreze et al. (1992) reported that temperature inversions (extremely strong stability), nearly ubiquitous over the ice-covered Arctic Ocean, tend to be surface-based from October through April, increasing in strength from October through winter in both depth and in the temperature difference from inversion base to top. For example, in October the median inversion depth is about 900m and the temperature difference is about 9K, whereas corresponding values in March are 1200 m and 12K. In summer, inversions are shallower and often elevated, with a deep mixed layer below. (There are also commonly shallow melt-induced surface-based inversions.) The seasonal cycle over Arctic land areas is similar but with temperature differences across the inversion of 14-16K (Figure 6).

Figure 7 shows a vertical cross section of potential temperature from the equator to 90°N for October. Potential temperature increases with altitude more steeply in the Arctic than at other latitudes, illustrating its stronger static stability. In turn, a larger vertical extent of warming in October compared to December would be expected given that stability increases from autumn into winter. In terms of potential temperature, at 80°N (for example) the increase in potential temperature from the surface to 850 hPa in October is 10K, versus 15K in December. From the surface to 700hPa, potential temperature increases by 20K in

October versus 25K in December. The atmosphere starts to cool freely to space at around 5-6 km above the surface (roughly

the 500 hPa level). While pronounced autumn warming does not extend upwards that far (Figure 5), the results nevertheless

argue that as amplified warming progresses, cooling to space will become more efficient as a negative feedback on autumn

warming.

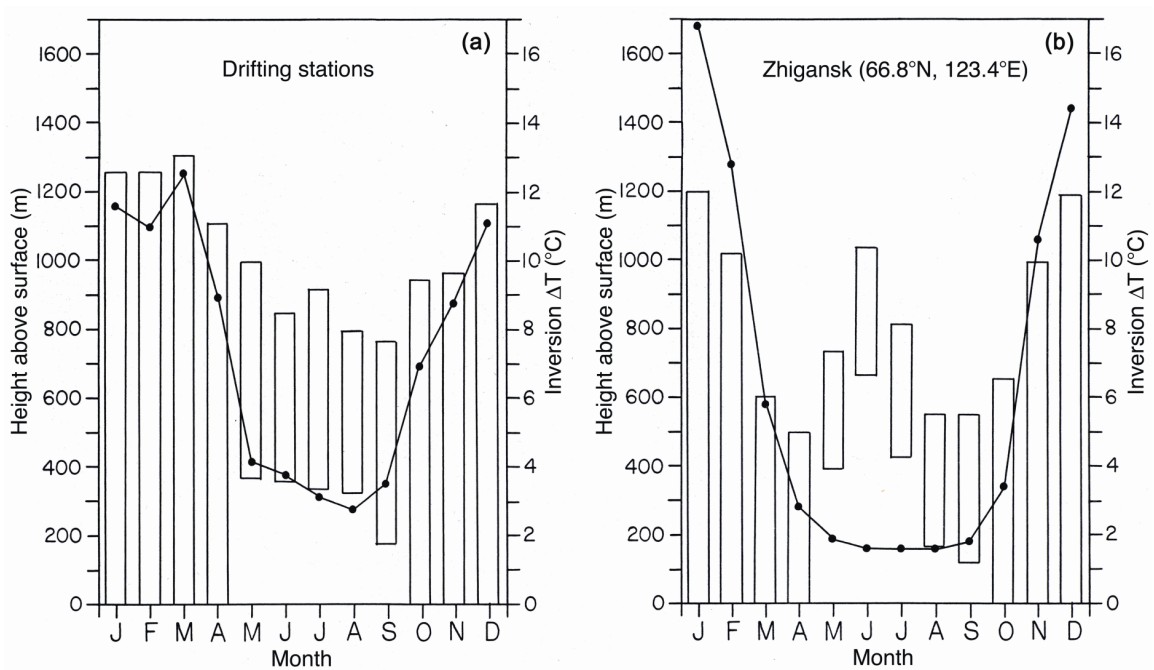

**Figure 6: Monthly median inversion top (top of bars), base (bottom of bars) and temperature difference (solid lines) from (a) drifting**

**station data from the central Arctic Ocean; (b) station Zhigansk over the Siberian tundra, taken as representative of the region**

**[from Serreze et al., 1992, by permission of AMS].**

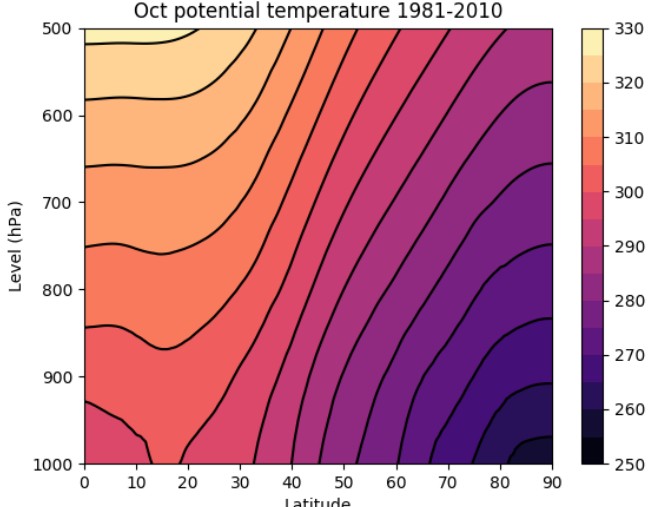

**Figure 7: Vertical cross section of zonally averaged potential temperature (K) from the equator to the pole for October, averaged over the period 1981-2010.**

**(a)** Oct Average dtheta/dp 1000-850 hPa 1980-2024      **(b)** Oct trend in dtheta/dp /year 1000-850 hPa 1980-2024

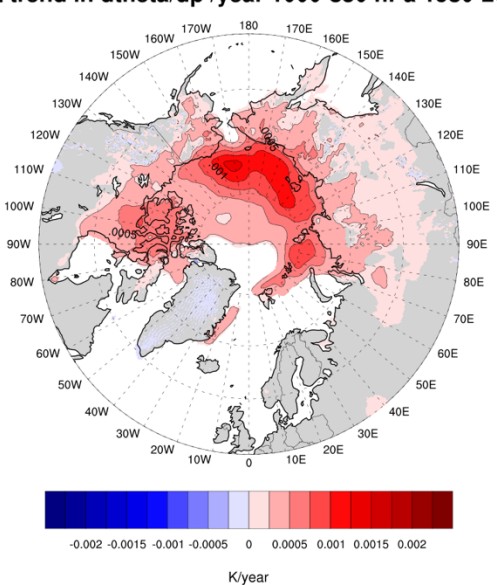

338

**Figure 8. Climatological averages (a, K/hPa) and linear trends (b, K/hPa per year) in low-level vertical stability (expressed as ($\theta_{850}$ - $\theta_{1000}$ ) / (850 hPa - 1000 hPa)) for October. Positive numbers for the climatological averages mean weaker stability, positive values for trends mean a decrease in stability with time. Only trends significant at $p<0.05$ are shaded based on an ordinary least squares regression test.**

Figure 8 shows climatological averages of surface to 850 hPa static stability for October, along with linear trends. In a stable atmosphere, dtheta/dP is negative (potential temperature increases with height while pressure decreases), so more negative values mean stronger stability. Consistent with Figure 7, there is a general increase in average stability moving polewards. However, stability is strongest north of Greenland and the Canadian Arctic Archipelago. It is likely not a coincidence that these areas have the thickest sea ice in the Arctic, implying especially small heat fluxes through the ice. Not surprisingly, large trends toward weaker static stability (positive values) dominate all the areas along the Eurasian coast, corresponding to the largest declines in September ice concentration, as well as in the Barents Sea, which has seen declines in winter. Smaller trends towards weaker stability dominate most of the rest of the Arctic Ocean, likely driven by a thinning ice pack. While the average conductive heat flux through most of the ice cover in October is on the order of 5-10 W $m^{-2}$ (upward), Liu and Zhang (2025) found that the conductive heat flux has increased since 1979 due to thinning, which outcompetes the effect of positive trends in surface skin temperatures. Our analysis finds support in the study of Simmonds and Li (2021) who find strong decreases in the Brunt–Vaisalla frequency over the Arctic and its broader region. We note here that the B-V frequency contains a 1/theta term which highlights the impact in the colder regions.

Corresponding results for December follow in Figure 9. Average stability is generally stronger than for October, with the clear exception of the Norwegian and Barents Seas and the extreme northern North Atlantic, where there is near neutral stability. The Norwegian and Barents Seas, in particular, have been recognized for unstable near-surface boundary layers in winter that develop during cold air outbreaks as Arctic air moves over open water surfaces, promoting strong surface heat fluxes and convective-type precipitation (Olaffson and Okland, 1994). Trends towards weaker stability are in turn prominent in the Barents Sea, the southern Chukchi Sea and Baffin and Hudson Bays, all areas where winter ice losses have been pronounced (especially the Barents Sea). Interesting in this regard is that weakening winter stratification may lead to intensification of near surface winds by increasing downward momentum transfer (Zapponini and Goessling, 2024), which will then foster stronger upward turbulent heat fluxes.

We stress that assessments of atmospheric stability and trends should be viewed with some caution. Based on comparisons with radiosonde profiles at coastal sites, Serreze et al. (2012) found that all three of the most modern reanalyses available at the time of that study (MERRA, NOAA CFSR, ERA-Interim) have positive cold-season temperature (and humidity) biases below the 850 hPa level and consequently did not capture observed low-level temperature and humidity and temperature inversions. MERRA had the smallest biases. Graham et al. (2019) similarly found a positive winter 2-m temperature bias in all six atmospheric reanalyses they compared to sea ice drifting stations – including ERA5. Additionally, Wang and Zhao (2024) found that the depiction of static stability over the Arctic in summer appears to be sensitive to the reanalysis product examined (ERA5, NCEP-R2 and JRA-55).

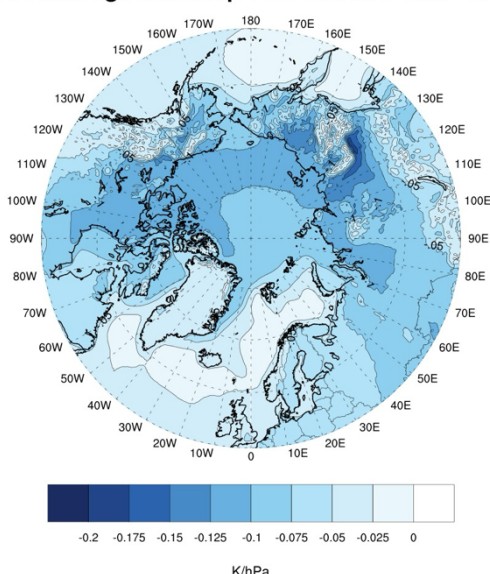
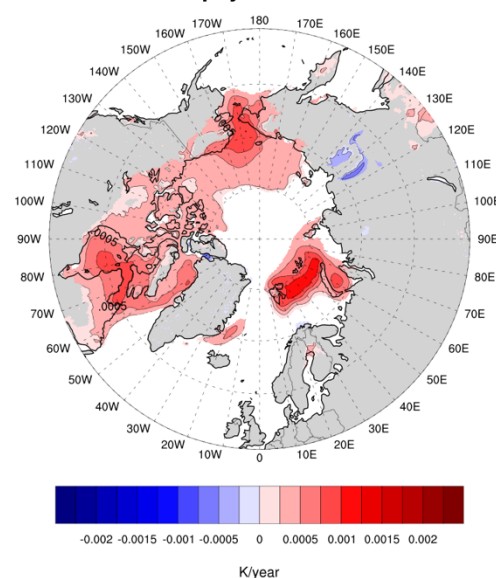

**(a) Dec Average dtheta/dp 1000-850 hPa 1980-2024**

**(b) Dec trend in dtheta/dp /year 1000-850 hPa 1980-2024**

373

**Figure 9. Climatological averages (a, K/hPa) and linear trends (b,K/hPa per year) in low-level vertical stability (expressed as ($\theta_{850}$ - $\theta_{1000}$ ) / (850 hPa - 1000 hPa )) for December.Positive numbers for the climatological averages mean weaker stability, positive values for trends mean a decrease in stability with time. Only trends significant at p<0.05 are shaded based on an ordinary least squares regression test.**

## 4 Discussion and Conclusions

The results presented here show a clear association between patterns of autumn and winter sea ice concentration trends and both the year-to-year evolution and seasonal expression of Arctic temperature anomalies. The link with sea ice loss can be viewed as an expression of seasonally delayed albedo feedback. We also see signals of variable atmospheric circulation in both temperature trends and the spatial structure of LAAs by decade. As discussed, a suite of other processes can also be linked to Arctic Amplification. Given that any process leading to warming will tend to enhance sea ice melt (spring and summer) or discourage its formation (autumn and winter), it can be viewed as serving to reinforce the key role of sea ice loss on observed AA.

Consider in this regard studies from coupled models showing that AA can arise without the albedo feedback through the lapse rate and Planck feedbacks (e.g., Caballero and Langen, 2005; Pithan and Mauritsen, 2014; Previdi et al., 2021). Lapse rate feedback relates to the stronger stability of the Arctic atmosphere compared to low latitudes, focusing the temperature rise closer to the surface and reducing longwave radiative cooling to space. From coupled simulations, Previdi et al. (2021) find that through positive lapse rate feedback, AA develops in only a few months following an instantaneous quadrupling of

atmospheric $CO_2$, well before any significant sea ice loss, although ice loss contributes significantly to warming after the first few months. While one can question what an instantaneous quadrupling of $CO_2$ teaches us about the real world, a key point is once sea ice begins to decline, the positive lapse rate feedback, keeping the heating near the surface, will contribute to spring and summer ice melt and delay seasonal ice growth. That static stability becomes stronger from autumn into winter indicates that focusing the heating near the surface will also be more effective in winter. Conversely, ice loss, and likely also heat fluxes, are changing the larger environment towards reduced stability at low levels.

Turning to the Planck feedback, the larger increase in Arctic temperatures required to bring the system back to radiative equilibrium in response to a forcing can also be seen as a process augmenting summer sea ice loss and delaying autumn and winter ice growth. Increased autumn cloud cover as a contributor to AA is closely tied to sea ice loss through reducing stability in the boundary layer, promoting large upward surface heat fluxes (e.g., Kay and Gettleman, 2012).

In parting, a key message stemming from the present study is that the process of AA must consider both its strong seasonality and that AA, which is generally assessed by comparing Arctic regional temperature trends against trends for the globe as a whole, comes about by the integration across the Arctic of large spatial heterogeneity of temperature changes, seen both in the spatial pattern of Arctic trends but especially when we look at the problem through local amplification anomalies – LAAs. While AA is small in summer, summer processes, namely the reduction of sea ice concentration and enhanced energy gain in the mixed layer, set the stage for the strong regional expressions of AA in autumn. These changes in spatial patterns of temperature anomalies extend into winter as areas of open water freeze over. In all seasons, variable atmospheric circulations appear to be important. Anomalous summer circulation can affect spatial patterns of September ice extent. In autumn and winter, these anomalous circulation patterns can affect temperature through advection as well as by their influence on sea ice concentration, such as in the Barents Sea. Static stability also changes seasonally, which will influence the vertical expression of temperature anomalies.

In short, the more we look at AA, the more we discover that it is a very complex beast. These complexities bear not only on the future evolution of AA and related impacts on permafrost warming and changes in the frequency of rain on snow events (Serreze et al., 2021), but on key issues such as potential impacts of Arctic warming on middle latitude weather patterns (Ding et al., 2024).

*Code and data availability:* The ERA5 data were obtained from the Research Data Archive at the National Center for Atmospheric Research: DOI: 10.5065/BH6N-5N20. Sea ice data was obtained from the National Snow and Ice Data Center https://nsidc.org/data/nsidc-0051/versions/2. For processing code contact Elizabeth Cassano (Elizabeth.Cassano@colorado.edu)

*Author contributions:* Mark Serreze wrote the first draft of the paper. Elizabeth Cassano performed the bulk of the data analysis and creation of figures and assisted in writing. Alex Crawford, John Cassano and Chen Zhang provided intellectual input to the paper and contributed to the writing.

*Competing interests.* The contact author has declared that none of the authors has any competing interests.

*Acknowledgement:* This study was supported by NSF Navigating the New Arctic Grant 1928230 and the Canada-150 Chair Program.

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
