# Peer review of "The Observed Evolution of Arctic Amplification over the Past"

_EGUsphere, 2025_

## Referee Comment (RC2)

**Review of EGUsphere-2025-3690**

The observed evolution of Arctic amplification over the past 45 years

by M. Serreze, E. Cassano, A. Crawford, J. Cassano, and C. Zhang

**Overview:** This study provides an update on the evolution of seasonal Arctic amplification (AA) during 1980 to 2024 using output mainly from ERA5. AA is calculated as differences between monthly mean Arctic two-meter air temperature anomalies (both pan-Arctic or individual Arctic gridpoints) and global-average two-meter air temperature anomalies. It should be noted this metric differs from that used in some other studies based on ratios of Arctic-to-global pace of temperature changes. While this manuscript provides no new revelations about Arctic amplification, it is a worthwhile addition as an update through 2024, especially the local amplification anomaly (LLA) metric that helps elucidate "hot spot" regions of AA in connection with sea-ice variability and horizontal advection. I don't have any major concerns or suggestions, and after addressing numerous minor suggestions/corrections/comments listed below, I recommend the manuscript be published.

**Specific comments and suggestions**

1. In many instances while reading the text, if found the tense confusing or awkward. Past events and conclusions from past papers were often described with the present tense when it seemed past tense was more appropriate. I request the authors reread the manuscript and decide which tense is indicated and be consistent throughout.

2. Add units to color bars in all plots. Labels on color bars are too small.

3. Line 59: Misplaced parenthesis

4. Figures 1, 2, 3, 4, 8, 9: I suggest removing some longitude labels (maybe every 10 or 20 degrees instead of 5?) to look less cluttered, and please add labels on some latitude lines.

5. Fig. 2: The plot for Sept is duplicated.

6. 110: "limiting" seems like an odd word to use here. Fluxes are always limited – maybe "reducing" is better?

7. 113: Please add a reference for thinning ice cover.

8. 116: Replace "of much of" to "over much of"

9. 137: Are the units of global anomalies in degrees per decade or per year? Please specify units in Table 1.

10. 139 and 141: Is Arctic defined as poleward of 50 or 60N?

11. Table 1, row 3: should be 2000-2009? Row 4 should be 2010-2019. Please specify units in table title.

12. 145: It would be valuable IMO to point out that AA can also enhance cooling, as presented in Table 1. Usually AA is understood as amplification of warming, but it can go both ways and has in the past.

13. 146: To increase clarity, I suggest beginning this sentence with: "During the decade at the middle of the baseline period…"

14. 157: "now" is confusing and extraneous here.

15. 165: It seems the comma after "values" should be a hyphen?

16. 187: "Regressed" has a specific statistical meaning, so I suggest replacing it with "decreased" for added clarity.

17. 193: Add "period" after 2010-2019

18. 193-194: "grow" appears twice in grown and growing

19. 197: I suggest adding "pan-Arctic" before "AA is somewhat smaller" if that is what is meant.

20. 204: Two "promotings" in this line

21. 227-228: Three "used" in this sentence

22. 233: I believe this should say "latitudes" 75-80N. Why is this latitude range so narrow? The zonal pattern of AA in SON and DJF is substantially wider.

23. 240: Although December anomalies are less vertically extensive, it may be worth noting they are likely to have a bigger impact on fluxes because of larger difference in temperature between the surface and air above.

24. 252: Perhaps note here that summer inversions are elevated? It seems contradictory to say summer inversions are shallower but accompanied by a deep mixed layer.

25. 256: I'm not sure this will be obvious to readers. Please explain how this plot illustrates variations in stability with latitude. Even better would be a plot of the vertical gradient in theta versus latitude.

26. 260: The decrease in cloud cover and moisture content from autumn into winter also tends to increase radiative cooling to space.

27. 263: Table 1 illustrates that AA can be negative, so perhaps "progresses" is not appropriate word here. Amplified warming might be clearer.

28. Figs. 7, 8, and 9 captions: add units.

29. Figs. 8 and 9: I suggest changing color scales to be just negative for left plots and just positive for right plots to more clearly display spatial variations.

30. 273 and 306: Units are unclear. "Over 1000-850" not clear – I believe it should be K/hPa. K/hPa is not a trend – what is unit of time? There's a typo at ends of these lines: "numbers in for the…"

31. 282: The word "trend" appears often in these lines – how about changing "downward trends" to "declines"?

32. 284: Remove one "October"

33. 285: Are these fluxes turbulent or just sensible? Do you mean upward fluxes have increased?

34. 290: Ditto.

35. 296-302: This information is pretty old (2012 paper). Maybe MERRA-2 is better? Please add more up-to-date information if it's available.

36. 315: I think "they" should be "it" to agree with "any process"

---

## Author Comment (AC1)

Please find below our responses to the reviewer comments and concerns regarding manuscript EGUsphere-2025-3690 "The observed evolution of Arctic amplification over the past 45 years" submitted to *The Cryosphere*. Our responses are provided in red.

Respectfully,

Mark C. Serreze, and co-authors

**REVIEWER #1**

**General Comments:**

This study provides an overview on the seasonal and long-term change in Arctic amplification (AA) over the last several decades using ERA5 reanalysis and satellite-derived sea ice data. The authors introduce a metric called the "Local Amplification Anomaly" (LAA), which is used to diagnose how Arctic near-surface temperatures are changing relative to the global average on a point-by-point basis. In addition to this, they look at changes in low-level stability over the Arctic, again through season and time. Overall, this study highlights the tightly coupled and complex set of relationships between sea ice melt/growth, upper ocean heat accumulation, and near-surface AA ratios, with a focus on the importance of considering these interactions depending on the season.

Overall, the methods and results are logical, and the paper is particularly well-written. The authors did a great job in describing AA in very clear and concise language. I just have some additional thoughts, questions, and comments below, which I hope are useful in the revision process.

We thank this reviewer for these positive comments and the time and effort spent on the review.

**Specific Comments:**

1. My primary concern is related to the originality/novelty of the results for publication in The Cryosphere (journal's reviewing criteria). At this stage, it is well understood now that the rate of AA differs by season across the Arctic, particularly because of different feedbacks like heat flux exchanges through the seasonal cycle of ice growth and melt. There have also been a few review papers on AA processes published in recent year (e.g., Henderson et al. 2021; Previdi et al. 2021; Taylor et al. 2022, Esau et al. 2023) and studies that have quantified an updated calculation of the AA ratio. I think there are a few new results nicely described here for the seasonal and decadal evolution of regional AA, so the authors should more clearly highlight the purpose and these new contributions in the introduction of the paper (such as around L44-54) relative to existing earlier work on Arctic change.

We thank the reviewer for bringing these papers to our attention. We have now briefly described and cited these papers in the introduction. The paper by Esau et al (2023) is particularly valuable in highlighting some of the impacts of AA on Arctic terrestrial and marine ecosystems. Additionally, we have also added text to point out what is new in our study and to set it apart from previous work.

2. This cautionary comment is acknowledged in a few places, such as L296-303, but I still have some hesitation around showing only results from ERA5. I understand that a comparison of observational/reanalysis datasets is not within the scope of this paper, but I still think it is necessary for a quick comparison for the temperature results given that ERA5 does have numerous biases (which could even be reflected in trend data) (Wang et al. 2019; Yu et al. 2021; Tian et al. 2024). One suggestion would be to show the seasonal decomposition of decadal AA, similar to Figures 3-4, in a supplement figure using a station-based gridded dataset (such as Berkeley Earth, or others).

Thank you for bringing these studies to our attention. We acknowledge that we were remiss in not adequately reviewing papers discussing bias in ERA5. We added the following text to Section 2 and return to the issue of biases later in the paper:

"Our results must be viewed within the context of known problems in ERA5, one being a warm bias in 2-meter air temperature over the Arctic (Yu et al., 2021; Tian et al., 2024). Compared to an extensive set of matching drifting observations, Yu et al. (2021) found ERA5 to have a mean bias of  $2.34 \pm 3.22$  °C in 2-meter air temperature, largest in April and smallest in September. Interestingly, surface (skin) temperature biases were found to be negative ( $-4.11 \pm 3.92$  °C overall), largest in December and smaller in the warmer months, although the magnitudes might be overestimated by the location of the surface temperature sensors on the buoys, which may have been affected by snow cover. While we are largely dealing in this paper with anomalies, rather than absolute values, our comparisons between Arctic and global anomalies may be influenced by the fact that biases at the global scale are different. Wang et al. (2019) found that compared to the earlier ERA-I effort, ERA5 has a larger warm bias at very low temperatures (<-25°C) but a smaller bias at higher temperatures. ERA5 has higher total precipitation and snowfall over Arctic sea ice. The snowpack in ERA5, results in less heat loss to the atmosphere and hence thinner ice at the end of the growth season, despite the warm bias."

In addition to adding this text highlighting biases in ERA5, we have performed the same analysis as done with ERA5 using the Berkeley Earth Surface Temperature (BEST) dataset. We have added verbiage in the paper around this analysis as well as adding supplemental figures and the analysis shown in Table 1 with BEST data. In Section 2, we added:

"To further address biases in ERA5, analysis was also performed using the 'Berkeley Earth Surface Temperatures' gridded surface temperature data (Rohde and Hausfather, 2020; Available for download from: <a href="https://berkeleyearth.org/data/">https://berkeleyearth.org/data/</a>). This dataset extends back to 1850, combining both 2m temperatures over land as well as sea surface temperatures to create a global, gridded observational dataset to which reanalysis data can be compared."

3. Changes in surface heat fluxes are mentioned throughout the study as supporting the evolution of events over time, including the association between near-surface air temperature, static stability, and sea-ice anomalies. I do think it would strengthen the results to actually show how turbulent heat fluxes unfold. If this additional analysis is not possible, please more clearly refer to other studies that have analyzed this data in observations/reanalysis.

Thank you for this suggestion. This analysis has been performed, and the results have been added to the paper.

**Technical Comments:**

1. L11; Spelling for "evolution"

Typo fixed.

2. L15; Spelling for "anomalies"

Typo fixed.

3. L59; Space in front of surface parameters.

Fixed.

4. L70-73; Is this point still actually under-acknowledged in the literature on Arctic change?

We have clarified as follows: "A key, but in our view, under-appreciated aspect of AA in our study is its strong seasonality - under-appreciated not that it exists but in the sense that processes at work during summer over the Arctic Ocean, when AA is small, set the stage for understanding the strong imprints of AA during autumn and winter."

5. L74-74; What is the statistical test used here and throughout the analysis?

The statistical test used was an ordinary least square regression. This has been added to the corresponding Figure captions.

6. L112; "Still[,] positive 2-meter"

Comma added.

7. L219-2020; Though this could be a product of internal variability and only looking at 5 years, compared the full decade.

The sentence starting "This is consistent with..." was meant to refer to the entire time series of mostly weaker anomalies, not that last 5 years. Therefore, we have flipped the order of these two sentences. That said, as the reviewer suggests, the importance of internal variability is obviously more prominent in 5 years than 10, so we also now mention that the 3°C anomalies for the 2020-2024 period may "represent short-term internal variability".

8. L227; "on" to "in"

Changed as suggested.

9. L228-230; Could a reference be added here? (e.g., <a href="https://nsidc.org/learn/ask-scientist/why-use-1981-2010-average-sea-ice">https://nsidc.org/learn/ask-scientist/why-use-1981-2010-average-sea-ice</a>)

The reference (Scott, M., 2022) has been added.

10. L289; Spelling for "Barents"

The "t" has been restored.

11. L336-338; Could this sentence be reworded to improve clarity?

This sentence has been reworded.

**Figures/Tables:**

All map Figures; Could the labels for the lines of longitude be reduced in frequency?
Sometimes the text overlaps and are difficult to read, along with too many dashed lines over the actual data contours.

This has been done.

2. All Figures; Could units and labels be added to the colorbars? This helps ensure greater accessibility for interpreting the results.

This has been done.

3. All Subplot Figures; Could letters be added (e.g., a, b, c, d) next to each subplot to make it easier to refer to the figures in the text?

This has been done.

4. Figure 1 and Elsewhere; Please define all acronyms (e.g., DJF, MAM, T2M)

This has been done.

5. Figure 2; I think it is showing September trends again on the right side.

The correct figure is in the paper now.

6. Table 1; Please indicate units.

Units have been added.

7. Figure 7; Could the colormap be changed here to a perpetually uniform, colorblind-friendly one that is more accessible to TC readers? (Hawkins, E. (2015). Scrap rainbow colour scales. Nature, 519(7543), 291-291.)

The figure has been remade with a more colorblind-friendly colormap.

**Reviewer's References:**

Esau, I., Pettersson, L. H., Cancet, M., Chapron, B., Chernokulsky, A., Donlon, C., ... & Johannesen, J. A. (2023). The arctic amplification and its impact: A synthesis through satellite observations. Remote Sensing, 15(5), 1354.

Henderson, G. R., Barrett, B. S., Wachowicz, L. J., Mattingly, K. S., Preece, J. R., & Mote, T. L. (2021). Local and remote atmospheric circulation drivers of Arctic change: A review. Frontiers in Earth Science, 9, 709896.

Previdi, M., Smith, K. L., & Polvani, L. M. (2021). Arctic amplification of climate change: a review of underlying mechanisms. Environmental Research Letters, 16(9), 093003.

Taylor, P. C., Boeke, R. C., Boisvert, L. N., Feldl, N., Henry, M., Huang, Y., ... & Tan, I. (2022). Process drivers, inter-model spread, and the path forward: A review of amplified Arctic warming. Frontiers in Earth Science, 9, 758361.

Tian, T., Yang, S., Høyer, J. L., Nielsen-Englyst, P., & Singha, S. (2024). Cooler Arctic surface temperatures simulated by climate models are closer to satellite-based data than the ERA5 reanalysis. Communications Earth & Environment, 5(1), 111.

Wang, C., Graham, R. M., Wang, K., Gerland, S., & Granskog, M. A. (2019). Comparison of ERA5 and ERA-Interim near-surface air temperature, snowfall and precipitation over Arctic sea ice: effects on sea ice thermodynamics and evolution. The Cryosphere, 13(6), 1661-1679.

Yu, Y., Xiao, W., Zhang, Z., Cheng, X., Hui, F., & Zhao, J. (2021). Evaluation of 2-m air temperature and surface temperature from ERA5 and ERA-I using buoy observations in the Arctic during 2010–2020. Remote Sensing, 13(14), 2813

Citation: https://doi.org/10.5194/egusphere-2025-3690-RC1

---

## Author Comment (AC2)

Please find below our responses to the reviewer comments and concerns regarding manuscript EGUsphere-2025-3690 "The observed evolution of Arctic amplification over the past 45 years" submitted to *The Cryosphere*. Our responses are provided in red.

Respectfully,

Mark C. Serreze, and co-authors

**REVIEWER #2**

Overview: This study provides an update on the evolution of seasonal Arctic amplification (AA) during 1980 to 2024 using output mainly from ERA5. AA is calculated as differences between monthly mean Arctic two-meter air temperature anomalies (both pan-Arctic or individual Arctic gridpoints) and global-average two-meter air temperature anomalies. It should be noted this metric differs from that used in some other studies based on ratios of Arctic-to-global pace of temperature changes. While this manuscript provides no new revelations about Arctic amplification, it is a worthwhile addition as an update through 2024, especially the local amplification anomaly (LLA) metric that helps elucidate "hot spot" regions of AA in connection with sea-ice variability and horizontal advection. I don't have any major concerns or suggestions, and after addressing numerous minor suggestions/corrections/comments listed below, I recommend the manuscript be published.

We appreciate the great time and effort invested in this review.

**Specific comments and suggestions**

In many instances while reading the text, if found the tense confusing or awkward. Past
events and conclusions from past papers were often described with the present tense
when it seemed past tense was more appropriate. I request the authors reread the
manuscript and decide which tense is indicated and be consistent throughout.

We have made references to past findings more consistent by switching any cases where we had been using present tense to either past tense or past perfect tense. We also added an additional two figure references and switch to more 1st person language when referring to our results. Collectively, we hope these changes improve the readability.

2. Add units to color bars in all plots. Labels on color bars are too small.

This has been done.

3. Line 59: Misplaced parenthesis

Corrected.

4. Figures 1, 2, 3, 4, 8, 9: I suggest removing some longitude labels (maybe every 10 or 20 degrees instead of 5?) to look less cluttered, and please add labels on some latitude lines.

This has been done.

5. Fig. 2: The plot for Sept is duplicated.

The correct plot is in the paper now.

6. 110: "limiting" seems like an odd word to use here. Fluxes are always limited – maybe "reducing" is better?

To make the phrasing less awkward, we change it to "...reducing energy transfer between the ocean and atmosphere."

7. 113: Please add a reference for thinning ice cover.

We have added two studies for this, one based on altimetry (Landy et al., 2022) and the other on moorings (Sumata et al., 2023).

- 8. 116: Replace "of much of" to "over much of" Replaced as suggested.
- 9. 137: Are the units of global anomalies in degrees per decade or per year? Please specify units in Table 1.

The units are simply degrees C since these are discrete anomalies, not trends. The units are now stated in the caption, as well as the reference period being 1981-2010.

10. 139 and 141: Is Arctic defined as poleward of 50 or 60N?

Including the region 50°-60°N in maps was done to provide additional context and comparison to higher mid-latitudes, but all spatially averaged values for "Arctic" are for poleward of 60°N. To make this distinction clearer, we have added the following statement at the end of the first results paragraph (new line 105).

"In this study, the Arctic is defined as areas poleward of 60°N, but maps extend down to 50°N to enable comparisons between changes in the Arctic and the higher middle latitudes."

11. Table 1, row 3: should be 2000-2009? Row 4 should be 2010-2019. Please specify units in table title.

Units are now specified and the row 3 and row 4 labels are fixed.

12. 145: It would be valuable IMO to point out that AA can also enhance cooling, as presented in Table 1. Usually AA is understood as amplification of warming, but it can go both ways and has in the past.

This is a tricky place to make that statement because we are looking at discrete anomalies relative to the 1981-2010 baseline, not a trend ratio. The temporal temperature trend for this period was still positive (still warming, in other words). Therefore, bringing up the point that AA goes both ways isn't super applicable here. However, the reviewer's point about Arctic amplification being applicable to both warming and cooling trends is apt. To avoid any confusion, then, we added this statement to the interpretation of Table 1.

"Note, since 1980-1989 is (primarily) the first decade of the 1981-2010 baseline period, greater negative anomalies for the Arctic than the globe still indicate amplified warming in the Arctic."

13. 146: To increase clarity, I suggest beginning this sentence with: "During the decade at the middle of the baseline period..."

The statement is written this way because, mathematically, the reason why 1990-1999 has the smallest anomalies is because it is the middle of a baseline period that contains a roughly linear trend. This should be clearly now based on the edits we made in response

to comments #10 and #11 from Reviewer 2 and comment #9 from Reviewer 1.

14. 157: "now" is confusing and extraneous here.

The word has been deleted.

15. 165: It seems the comma after "values" should be a hyphen?

Yes, indeed. It has been changed.

16. 187: "Regressed" has a specific statistical meaning, so I suggest replacing it with "decreased" for added clarity.

Changed as suggested.

17. 193: Add "period" after 2010-2019

This word has been added.

18. 193-194: "grow" appears twice in grown and growing

The first instance has been replaced with "intensified."

19. 197: I suggest adding "pan-Arctic" before "AA is somewhat smaller" if that is what is meant.

This has been added.

20. 204: Two "promotings" in this line

The first instance has been replaced with "facilitating".

21. 227-228: Three "used" in this sentence

We changed the first instance to "applied". Another edit (based on a comment by reviewer 1), eliminated the second.

22. 233: I believe this should say "latitudes" 75-80N. Why is this latitude range so narrow? The zonal pattern of AA in SON and DJF is substantially wider.

Yes, that should say "latitudes", which has been fixed. The 75°-80°N range was selected because it represents the latitude band where the air temperature anomalies are most constantly pronounced across both SON and DJF. Although the zonal pattern of Arctic amplification extends more broadly (from about 70°N to 85°N depending on region and season).

23. 240: Although December anomalies are less vertically extensive, it may be worth noting they are likely to have a bigger impact on fluxes because of larger difference in temperature between the surface and air above.

We thank the reviewer for this thoughtful observation. While Figure 5 shows temperature anomalies rather than absolute temperatures and therefore does not allow a direct assessment of the vertical temperature gradient, we agree that it is valid point and that such anomalies could potentially enhance surface fluxes. We have added a brief statement to acknowledge this potential effect.

"Although these anomalies are less vertically extensive, the stronger near-surface temperature difference between the surface and the air above in December could potentially enhance surface fluxes".

24. 252: Perhaps note here that summer inversions are elevated? It seems contradictory to say summer inversions are shallower but accompanied by a deep mixed layer.

Summer inversions are now described as often being elevated as well.

25. 256: I'm not sure this will be obvious to readers. Please explain how this plot illustrates variations in stability with latitude. Even better would be a plot of the vertical gradient in theta versus latitude.

To clarify, we have replaced the original sentence with: "Potential temperature increases with altitude more steeply in the Arctic than at lower latitudes, illustrating its, the stronger the static stability."

26. 260: The decrease in cloud cover and moisture content from autumn into winter also tends to increase radiative cooling to space.

Our writing here was imprecise. The reviewer is of course right about the radiative cooling being more efficient when less clouds/moisture means less of a greenhouse effect. What we were trying to describe was the Planck and lapse rate feedbacks on the trends, not the mean state. We now reference Figure 5 and rephrased the statement to improve the clarity:

"While pronounced autumn warming does not extend upwards that far (Figure 5), the results nevertheless argue that as amplified warming progresses, cooling to space will become more efficient as a negative feedback on autumn warming."

27. 263: Table 1 illustrates that AA can be negative, so perhaps "progresses" is not appropriate word here. Amplified warming might be clearer.

Changed as suggested.

28. Figs. 7, 8, and 9 captions: add units.

We have added units to all three captions.

29. Figs. 8 and 9: I suggest changing color scales to be just negative for left plots and just positive for right plots to more clearly display spatial variations.

The scale has been changed for the left plot for clarity but kept the same for the right plot given that there are (albeit very small) areas of negative values.

30. 273 and 306: Units are unclear. "Over 1000-850" not clear – I believe it should be K/hPa. K/hPa is not a trend – what is unit of time? There's a typo at ends of these lines: "numbers in for the..."

The captions for Figures 8 and 9 have been revised. The units are now explicit for both the averages and the trends, and the formula for calculating stability (difference in potential temperature divided by difference in pressure) is now expressed mathematically as " $(\vartheta_{850} - \vartheta_{1000})/(850 \text{ hPa} - 1000 \text{ hPa})$ "

31. 282: The word "trend" appears often in these lines – how about changing "downward trends" to "declines"?

Two instances of "downward trends" were switched to "declines" to vary the language.

32. 284: Remove one "October"

The first instance was removed.

33. 285: Are these fluxes turbulent or just sensible? Do you mean upward fluxes have increased?

The flux described here is the conductive flux through the sea ice medium, not an atmospheric turbulent flux. To clarify, we have changed from the plural "conductive heat fluxes" to the singular "conductive heat flux". We also now make clear that the average conductive flux through sea ice is directed upward in October. Finally, we switch "upwards trends" to "positive trends" to avoid confusion about the meaning of the word "upward".

34. 290: Ditto.

See previous comment – several tweaks were made to clarify.

35. 296-302: This information is pretty old (2012 paper). Maybe MERRA-2 is better? Please add more up-to-date information if it's available.

We were able to improve this discussion by citing a more recent reanalysis validation study (Graham et al., 2019) that includes both MERRA-2 and ERA5 (and therefore show the warm bias in the near-surface atmosphere in the winter persists).

36. 315: I think "they" should be "it" to agree with "any process" Fixed.

---

## Author Comment (AC3)

Please find below our responses to the reviewer comments and concerns regarding manuscript EGUsphere-2025-3690 "The observed evolution of Arctic amplification over the past 45 years" submitted to *The Cryosphere*. Our responses are provided in red.

Respectfully,

Mark C. Serreze, and co-authors

**REVIEWER #3**

This is an interesting paper which explores the regional (or local) aspects of AA. Many studies will refer to the nature or impacts of Arctic change as a whole, but this manuscript delves a little deeper into the issue. The submission has the potential to make a significant contribution to the literature, but it is not quite there yet. Before I would be able to recommend acceptance, there are a number of issues which need to be addressed.

We thank the reviewer for this positive review and for their time and effort.

Lines 24 - : In this introductory survey and remarks it would be appropriate to mention the 'PAMIP' project ...

Doug M. Smith, James A. Screen, Clara Deser, Judah Cohen, John C. Fyfe, Javier Garcia-Serrano, Thomas Jung, Vladimir Kattsov, ... and Xiangdong Zhang, 2019: The Polar Amplification Model Intercomparison Project (PAMIP) contribution to CMIP6: Investigating the causes and consequences of polar amplification. *Geoscientific Model Development*, **12**, 1139-1164, doi: 10.5194/gmd-12-1139-2019.

Thank you for the suggestions. We have included this work in the introduction as: "The Polar Amplification Model Intercomparison Project (PAMIP; Smith et al., 2019) further investigates the causes and consequences of polar amplification using a coordinated set of numerical model experiments, providing valuable insights into the mechanisms driving AA."

Line 39: 'Graversen et al.' should be 'Graversen and Burtu'.

Fixed as suggested.

Line 45: Important additional rationale for this work is that the local characteristics of AA have broader implications. Beneficial here to support this by referencing study of Wenqin Zhuo, Yao Yao & co-authors, 2023: The key atmospheric drivers linking regional Arctic amplification with East Asian cold extremes. Atmosp. Res, 283, 106557, doi: 10.1016/j.atmosres.2022.106557 who demonstrate the AA regionality is important in producing very different remote influences, via teleconnection patterns, into the midlatitudes.

We thank the reviewer for this helpful suggestion. We have added a sentence to highlight the broader implications of regional Arctic amplification, citing Zhuo et al., (2023):

"The local characteristics of AA are important, as regional variations can produce different

remote influences, including midlatitude climate extremes (Zhou et al., 2023)."

Line 73: Better to write as '2' and '5'.

Words have been changed to numbers here.

Line 80 (Figures): Showing the values of longitude at every 5 degrees makes these plots look unnecessarily busy. Much less frequent would be ample.

Also the headers on the plots should read 'trend' and not 'change'

The caption reads ... 'Shading is used for trends significant at p<0.05'. This is confusing, especially as in the caption of Figure 2 the authors have (a better expression of) 'Only trends significant at p<0.05 are shaded'. Be consistent and as clear as possible.

The preferred phrasing is now used for all figures with trends (Figure 1, 2, 8, and 9). The figures have been redone with the frequency of longitude labeling reduced, and the heading on the plots read 'trend' now rather than change.

Line 85: The authors must explain how they performed the statistical significance test. An additional aspect on this is that the parameters discussed here have considerable memory (autocorrelation). This has the effect of reducing the 'effective' number of data points and hence reduces to degrees of freedom. Please to allow for this also – see approach of Christopher S. Bretherton, Martin Widmann, Valentin P. Dymnikov, John M. Wallace and Ileana Bladé, 1999: The effective number of spatial degrees of freedom of a time-varying field. Journal of Climate, 12, 1990-2009, doi: 10.1175/1520-0442(1999)012<1990:TENOSD>2.0.CO;2.

We thank the reviewer for this comment. The statistical significance test for linear trends in sea ice concentration (% per decade) was performed using ordinary least squares regression test, and only trends significant at p < 0.05 are shaded for each grid point. While year-to-year autocorrelation in annual anomalies is minimal, spatial autocorrelation is present because neighboring grid points might not be independent. To account for this in a practical way, we focus on large contiguous areas with robust effect sizes, rather than isolated small pockets of significance, which are more likely influenced by spatial correlation. This approach emphasizes physically meaningful patterns while mitigating the influence of spatial autocorrelation, and we have clarified this in the Methods section.

Line 143: The third and fourth entries into the first column of Table 1 should be '2000-2009' and '2010-2019'.

This has been corrected.

Lines 175-180: An interesting argument is made here. Note that in the three later decades shown in Figure 4 the AA over Eurasia is prominently negative. This ties in neatly with the Warm arctic-cold Eurasia (WACE) phenomenon (refence here Li, M., et al., 2021: Anchoring of atmospheric teleconnection patterns by Arctic Sea ice loss and its link to winter cold anomalies in East Asia. *Int. J. Climatol.*, **41**, 547–558). In line with the authors' comments here regarding

the impact of the phase of the North Atlantic Oscillation in the earlier period, studies have shown that in more recent times other large-scale modes (such as Interdecadal Pacific Oscillation and the Atlantic Multidecadal Oscillation) have influenced the nature of the WACE pattern (see ...

Luo, et al., 2022: The modulation of Interdecadal Pacific Oscillation and Atlantic Multidecadal Oscillation on winter Eurasian cold anomaly via the Ural blocking change. *Climate Dyn.*, doi: 10.1007/s00382-021-06119-7 and

Luo, B., D. Luo, and coauthors, 2022: Decadal variability of winter warm Arctic-cold Eurasia dipole patterns modulated by Pacific Decadal Oscillation and Atlantic Multidecadal Oscillation. *Earth's Future*, **10**, e2021EF002351, doi: 10.1029/2021EF002351). The paper will benefit from a more incisive argument along these lines on the structure of the Fig. 4 plots.

Thank you for pointing this out. We have added the following to the text: "The observation that the last three time periods have negative LAA values over Eurasia is of interest through its apparent link with Warm Arctic-Cold Eurasia (WACE) phenomenon-while AA has become increasingly prominent, this has been attended by recent surface cooling over Eurasia, most evident in winter with considerable decadal variability. (e.g., Gong et al., 2017). The WACE phenomenon has garnered considerable attention over the past decades, and a suite of driving factors has been offered. An Urals blocking pattern has been identified as playing a strong role, and recent work has shown that decadal variability in the WACE phenomenon is mediated by phases of the Pacific Oscillation and the Atlantic Multidecadal Oscillation (e.g., Luo et al., 2022)."

Line 183 (Figure 3): Figs. 3 and 4 present much more information than does Table 1. Perhaps consider deleting the Table as it contains lots of number. If you follow that, maybe also here show LAA Figs. covering spring and summer.

Also suggest 'LAA' rather than 'AA' in the headings of the sub-plots.

We will have to agree to disagree on this point; we feel that Table 3 provides valuable information. Note that changes have been made to Table 3 in response to other reviewer comments. However, the headings in the plots have been changed from 'AA' to 'LAA'.

Line 207: Support this point by also referencing Lee S, Gong T et al. (2017) Revisiting the cause of the 1989-2009 Arctic surface warming using the surface energy budget: Downward infrared radiation dominates the surface fluxes. Geophys. Res. Lett. 44: 10,654–10,661 doi: 10.1002/2017GL075375.

We have added this reference.

Lines 234-5: Why just October and December, when up till now you have been considering SON and DJF?

October is when there are particularly large heat fluxes from the ocean to atmosphere, while in December, most of these areas (apart from the Barents Sea) have re-frozen. We wanted to capture that contrast. We now note this in the text.

Line 251: Paper has been using 'deg C' up to here, and now 'K'. Please revert to deg C here in the subsequent occurrences.

Everything has been changed to Kelvin for consistency.

Line 271 (Heading on Figures): Please to change 'mb' to 'hPa'. (Similar for Figure 9).

This has been done.

Lines 276-286: reinforce these arguments by referencing paper of Simmonds et al. (2021 - Trends and variability in polar sea ice, global atmospheric circulations, and baroclinicity *Ann. NY Acad. Sci.* **1504** 167-86) showing strong decreases in the Brunt–Vaisalla frequency over the Artic and its broader region.

On this issue the B-V frequency is more strongly connected to the (thermo)dynamics than is 'delta theta'. It also contains a '1/theta' term which highlights the impact in the colder regions. Some words on this are warranted here.

Done as suggested. Thank you for making us aware of this study.

Lines 302-303: Valuable to mention in text here that Xin Wang and Jinping Zhao used three data sets, namely NCEP-R2, ERA5, and JRA-55, to make it explicit that ERA5 (as used here) was one of the sets.

Done.

Lines 412-413: Please to include full bibliographic details (volume, article number, ...) here ...

Liu Y, Zhang J (2025) Conductive heat flux over Arctic sea ice from 1979 to 2022. J. Geophys. Res. 130: e2024JC022062 doi: 10.1029/2024JC022062.

Full details added.

Lines 446-7: Reference is repeated. From context, I suspect authors meant to make Screen JA, Simmonds I (2010) Increasing fall-winter energy loss from the Arctic Ocean and its role in Arctic temperature amplification. Geophys. Res. Lett. 37: L16707 doi: 10.1029/2010GL044136 as 'part b'.

The extra reference is now removed, and the "part b" version is now the only version.

More missing details in References ...

Stroeve, J. C., Markus, T., Boisvert, L., Miller, J. and Barrett, A. 2014. 'Changes in Arctic melt season and implications for sea ice loss', *Geophys. Res. Lett.* **41**, 1216-1225, doi: 10.1002/2013gl058951,

Stroeve, J. and Notz, D. 2018. Changing state of Arctic sea ice across all seasons. Env. Res. Lett. 13, 103001. DOI: 10.1088/1748-9326/aade56. ...

Please to check all reference informations carefully.

These two errors are fixed, and we went through the rest of the reference list, cleaning up a few more typos and omissions of details.